# Mineral surface chemistry control for origin of prebiotic peptides

Valentina Erastova [1], Matteo T. Degiacomi [1], Donald G. Fraser [2] & H. Chris Greenwell [3]

Some seventy years ago, John Desmond Bernal proposed a role for clays in the origin of life. While much research has since been dedicated to the study of silicate clays, layered double hydroxides, believed to be common on the early Earth, have received only limited attention. Here we examine the role that layered hydroxides could have played in prebiotic peptide formation. We demonstrate how these minerals can concentrate, align and act as adsorption templates for amino acids, and during wetting—drying cycles, promote peptide bond formation. This enables us to propose a testable mechanism for the growth of peptides at layered double hydroxide interfaces in an early Earth environment. Our results provide insights into the potential role of mineral surfaces in mimicking aspects of biochemical reaction pathways.

[1] Department of Chemistry, Durham University, South Road, Durham DH1 3LE, UK. [2] Department of Earth Sciences, University of Oxford, South Parks Road, Oxford OX1 3AN, UK. [3] Department of Earth Sciences, Durham University, South Road, Durham DH1 3LE, UK. Valentina Erastova and Matteo T. Degiacomi contributed equally to this work. Correspondence and requests for materials should be addressed to V.E. (email: valentina.erastova@durham.ac.uk) or to H.C.G. (email: chris.greenwell@durham.ac.uk)

The emergence of life, the point of transition from organic geochemistry to biochemistry, is one of the enduring unresolved questions in science. Though even defining life is still a matter of debate, the abilities to metabolize and replicate with inheritable mutations are essential criteria for Darwinian evolution. Life as we know it today, is dependent on the fidelity of information transfer through sequential polymeric systems, i.e. nucleic acids and peptides, interconverting structure into information, and information into structure.

Following Wöhler's abiological synthesis of urea, many hypotheses have been proposed for the origin of life through molecular evolution, in which simple organic molecules, whether they be amino acids, carbohydrates, or nucleosides, are initially formed from simple abiotic reactants. This synthesis of simple starting components is followed by the subsequent polymerization of organic monomers into biomolecules of increasing complexity and function. It is the latter step into which we seek to add insight by this present study. A number of challenges arise when attempting to understand how proto-biological monomers can form oligomers, which in time may become both functional and capable of Darwinian evolution. In living systems, biological catalysts and enzymes fill this role working through a variety of mechanisms based on the active sites and tertiary structures formed by proteins.

The idea of using hydrated mineral surfaces to replace biological catalysts on a pre-biological Earth dates to the 1940s, following Oparin and Bernal[1, 2]. Such surfaces offer sites at which simple monomers can concentrate from dilute solutions. Some of these surfaces may have very high enthalpies of rehydration, providing a driving force for condensation reactions. As such, hydroxides, silicates, carbonates, and borates have all been studied as reagents capable of aiding the polymerization of prebiotic monomers[3]. The internal surface of minerals, whether pores in three-dimensional systems or interlayer regions in two-dimensional systems, also offers a safe haven for nascent biopolymers from the effects of UV radiation[4].

Amino acids provide an interesting starting point for studying mineral—biomolecule interactions[5]. Protein—mineral interactions are prevalent in many biomineralization pathways, as well as are of interest to early Earth chemistry. Unlike nucleic acids for which a plausible prebiotic synthesis route is still being developed[6], the presence of amino acids on the Hadean Earth is considered very likely. Amino acids have been found in meteorites and other cosmic bodies[7, 8], which indicates a simple chemical synthesis process in the interstellar medium. Experimental evidence is also available for the abiotic synthesis of amino acids, under a variety of potential early Earth environmental conditions[9–12]. The charge of amino acids is pH-dependent and, hence, allows their association with different minerals in different environments.

A potential drawback of the mineral catalyzed synthesis of biopolymers is that polymers, with multiple points of attachment to a mineral surface, can be inherently hard to remove; as Lambert[3] noted, "If the initial steps of life really occurred on surfaces, how then did life escape surfaces at a later stage?"; thus, a full transition to biopolymers would require a significant change in external conditions to remove the polymer (or dissolve the mineral). An additional hurdle is that, whereas many negatively charged biomolecules exist, there are relatively few mineral surfaces with net permanent positive charge. The most common hydrated aluminosilicate minerals all carry net negative layer charges owing to permanent isomorphous substitutions. In order to act as catalytic templates for negatively charged biomolecules (i.e. nucleic acids or proteins at pH > 7), charge inversion is required through, for example, bridging cations.

One of the challenges in studying early Earth biomolecule evolution pathways is thus, to identify plausible environments and conditions for the occurrence of pre-biotic chemistry. Following the seminal work of Russell and Martin, low to mid-temperature alkaline hydrothermal systems have been shown to be suitable[13, 14]. Present-day analogs such as, the lost city hydrothermal vent field (LCHF) have also attracted much research interest since their discovery in 2000[15]. The LCHF is believed to have been active for 30,000 years. More recently, Price has been investigating the Strytan Hydrothermal Field, a Lost City-like Hydrothermal Vent in shallow waters. These groups of vents have high pH of 9–11, and temperatures of 70–150 °C. The vents are powered by exothermic serpentinization reactions. Owing to their likely ancient existence on Earth, their exothermic nature and presence of layered ordered inorganic materials, these vents could be a plausible location for life's origin.

Many studies have been carried out on silicate clays, to study their potential role in the formation of protobiomolecules[3, 16–19]. In contrast, layered double hydroxides (LDH), which also exist in hydrothermal vents and were common in early Earth[20], have attracted only limited attention[21, 22]. LDH materials are mixed brucite like clays with positive layer charges, created by the substitution of 2+ with 3+ metal ions. These positive charge sites give rise to ion exchange properties, allowing LDHs to concentrate amino acids, and act as templates for polymerization, as well as protecting reaction products.

Characterization of the complexes between small monomers and hydrated mineral surfaces is difficult, especially when the interactions occur in internal pores or between the layers of clay or clay like minerals. Addressing this area has shown the combination of methods in computational and experimental chemistry, with the former providing atomic and molecular level insight of mineral—organic interactions. Molecular dynamics simulations have notably been employed by Coveney and co-workers to study the interactions of nucleic acids with alumino-silicates and LDHs[23, 24]. Whereas nucleic acid—surface interactions have been studied in some detail, there remains a remarkable paucity of simulation data for peptide—LDH interactions.

We carry out a large scale computational modeling study of interactions between amino acids and LDHs under reducing early Earth conditions. Our LDH layers have the composition of $[Mg_3Al(OH)_8]^+$, while interacting with amino acids and peptides are deprotonated at pH 9.5. For this study, we chose a variety of amino acids (alanine, aspartate, leucine, lysine, histidine, and tyrosine), their mixtures, di- and hexa- peptides, and a randomly created 24-amino-acid-long peptides from the amino acid distributions mimicking naturally occurring ones. Analysis of diffusion, adsorption and arrangement of amino acids onto the LDH surface, as well as their concentration dependence and trends in a wetting—drying cycle, reveals possible mechanisms for peptide formation.

## Results

**Intercalation of amino acids affects LDH layer dynamics**. Irrespective of the identity of the amino acid, the LDH interlayer dehydrates with the same trend, indicating that the basal $d$-spacing is proportional to the number of atoms (organic load, Fig. 1) present in the interlayer, rather than the charge on amino acids (Supplementary Fig. 2a). Moreover the distance between two adjacent LDH layers is not constant across an interlayer, because the LDH layers display undulations (Supplementary Fig. 2b) caused by an aggregation of amino acids that locally bridge the layers, thus reducing the local $d$-spacing. The water expelled from these regions leads to swelling of neighboring areas. This

peristaltic-like phenomenon is particularly apparent in the case of leucine and tyrosine at 15 waters per amino acid (W/AA). Leucine has a large and strongly hydrophobic side chain, which allows leucine molecules that are adsorbed onto opposing LDH layer faces to interact, thus pulling the layers together. Where no such interaction is possible, the expelled water aggregates, and then leucine molecules rearrange, pointing their hydrophilic C-terminals towards the accumulated interlayer water to shield their hydrophobic side-chains. In the case of tyrosine, undulations arise from an interaction between an OH group of the side-chain of one amino acid, and a C-terminal of the other. This is due to the competition between the hydroxyl groups of the LDH and tyrosine. At 10 W/AA both tyrosine and leucine side-chains are long enough to interlock with the opposing ones, making the d-spacing constant with low-amplitude undulations. In the case of aspartate, at 20 W/AA (i.e. 10 W/Al) some amino acids are able to bridge two layers via a mediating molecule of water. This again brings the layers together, creating larger undulations. At 15 W/AA nearly all aspartate molecules are able to bridge across to the opposing layer either via bridging water or directly, creating evenly layered LDHs.

**Amino acids and peptides adsorb on LDHs via their C-termini.** All systems, except lysine, show a constant increase in amino acid adsorption on the LDH surface (Fig. 2a) upon dehydration. At hydration greater than 7 W/AA, approximately 75% of amino acids are adsorbed, with nearly all amino acids adsorbed at lower hydrations. At higher hydration levels aspartate shows preferential adsorption via its backbone than its side-chain (by a factor of 1.3). This is because the carbon in the C-terminal has a lower positive partial charge (0.34) than the carbon in the side-chain carboxylate (0.62). Notably, some aspartate molecules adsorb via both the side-chain and the backbone. Below 10 W/AA both backbone and side-chain carboxylates are fully adsorbed. Zwitterionic lysine shows a steady increase of adsorption from 30% at the highest hydration towards 90% at no interlayer water. Here the side-chain is strongly positive, and therefore interacts with the carboxylate of the backbone of neighboring molecules, reducing the adsorption on the LDH surface. In the case of peptides, shorter chains show higher adsorption than longer ones (Fig. 2b). In the case of aspartate, there is still a preference for adsorption via the backbone than by a side-chain factor of 1.3. Importantly, simulations of peptide mixtures (MIX3 and MIX4) indicate that further increase in chain length does not lead to a substantial decrease in their adsorption.

**Arrangement of adsorbed species is templated by LDH structure.** In order to detect possible templating effects of the LDH on the arrangement of adsorbed amino acids, we analyzed the radial distribution function (RDF) of amino acids' C-terminal oxygen atoms with respect to LDH's aluminum (Supplementary Fig. 3). In both the cases of pure systems and amino acid mixtures (MIX1 and MIX2), peaks matching the distribution of aluminum atoms are observed, indicating that the arrangement of amino acids' C-termini onto the LDH surface is templated by LDH charging sites.

To gain further insights into the arrangement of amino acids, we analyzed their orientation with respect to the surface. At high levels of hydration, both amino acids and peptides mostly adsorb by their C-termini. Upon dehydration, backbones uniformly arrange, so that C-termini oxygen atoms either adsorb on the same surface or bridge between two adjacent ones (Supplementary Fig. 4a, b). These observations indicate that dehydration enforces specific alignment on adsorbed species.

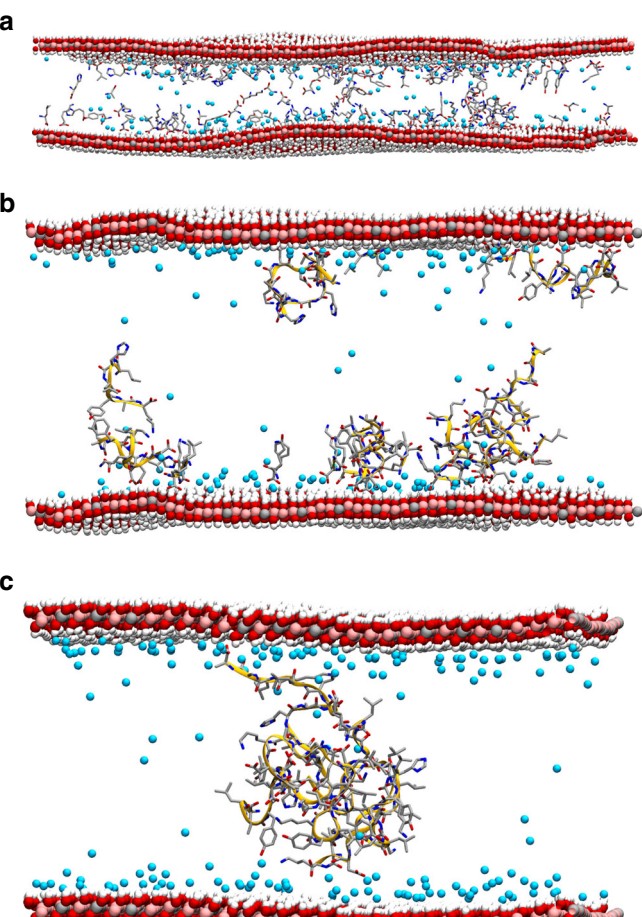

**Fig. 1** Amino acids and peptides intercalated in hydrated LDH. Example of modeled systems, showing how natural mixtures of **a** amino acids, **b** short, and **c** long peptides adsorb onto the LDH interlayers via their C-terminal. Colors are as follows: Mg, pink spheres; Al, gray spheres; Cl, blue spheres; O, red; H, white; N, blue; and backbone is represented with a yellow spline. For clarity water and H atoms on the amino acids are not shown

**LDHs promote amino acid polymerization.** Deprotonated amino acids hold a strongly negative charge on the carboxyl side that prevents nucleophilic attack by the amino group. Our quantum mechanical calculations show that amino acid adsorption onto the LDH surface allows redistribution of charges (from −0.41e to −0.23e on oxygen and 0.07e to 0.18e on carbon), thus activating the carboxylic group for subsequent peptide bond formation (charges given in Supplementary Table 2 and Supplementary Fig. 7).

When concentrated on the LDH surface, amino acids are prone to co-alignment. Figure 3b highlights close contacts between C- and N-termini. Upon dehydration C-termini are observed to interact with the LDH in multiple ways, all providing an accessible site for potential polymerization (Fig. 3c). We quantified the amount of these sites as a function of hydration, obtaining a count of reactive pairs. While hydrated systems (over 7 W/AA) only feature a small number of pairs (below 5%, Fig. 3a) dehydration leads to their rapid increase. In the case of tyrosine and lysine this phenomenon is only modest, because of competitive interactions between C-termini and $NH_3^+$ (lysine) or OH (tyrosine) groups of the side-chains. In the case of leucine, histidine, and alanine, the increase is more significant with up to 25% of amino acids coordinating in a potentially reactive arrangement upon dehydration. Alanine shows the highest

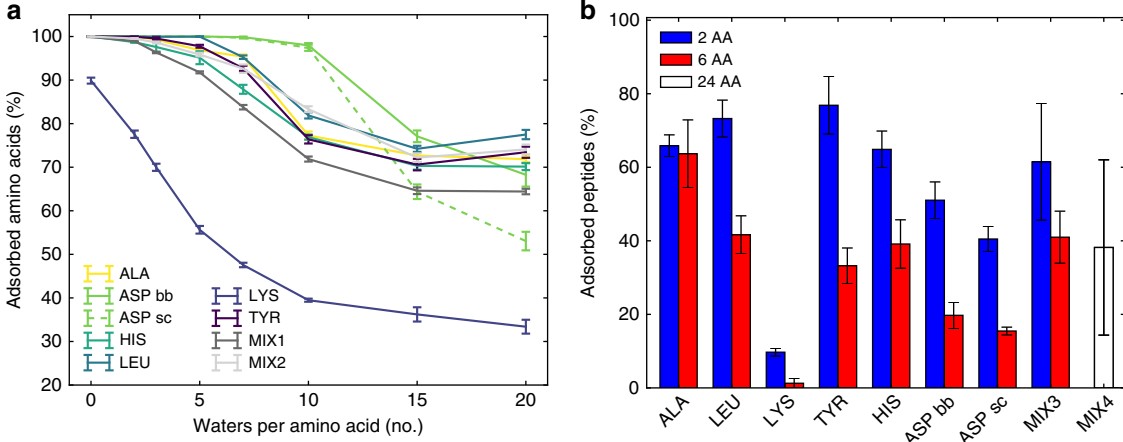

**Fig. 2** Amino acids and peptides adsorbed onto LDH. **a** Percentage of adsorbed amino acids as a function of their hydration (number of water molecules per amino acid). **b** Percentage of adsorbed di-peptides, hexa-peptides, and 24-mers in a fully hydrated system. Error bars represent standard deviation. With the exception of lysine, both single amino acids and peptides adsorb onto the LDH surface (distance <2.5 Å). Aspartate can adsorb both via its backbone (labeled ASP bb) and side-chain (ASP sc). Though less adsorbed than di-peptides, hexa-peptides, and 24-mers still feature significant adsorption rates, indicating that chain length has only a minor effect on peptide adsorption

increase in reactive pairs, as its small side-chain does not hinder its backbone co-alignment. In the case of aspartate, the highest number of reactive pairs occurs at 2 W/AA, but, unlike other amino acids, no further increase is observed upon full dehydration because of its lower concentration per unit volume (due to its double charge). Aspartate can form α- or β-peptides and in this work we report the sum of two. Importantly, mixed amino acids behave similarly to pure systems, and at 40% load (MIX2) the amount of reactive pairs is still comparable to that of a fully saturated system (MIX1). This indicates that the process of reactive pair formation is mostly independent from both amino acids' nature and saturation of LDH interlayers.

**Peptide bond formation on LDHs is energetically favorable**. All of the systems readily rehydrate when exposed to water (Supplementary Fig. 5), as their hydration energy is always smaller than that of a reference SPC water system ($-33.25$ kJ mol$^{-1}$). Such behavior is only slightly dependent on the nature of the intercalated amino acids. The more the systems dehydrate, the higher the energy gain. The formation of a peptide bond releases a molecule of water, thus contributing to the rehydration of the interlayer. We note that peptide bond formation is endergonic with a free energy change of 10–20 kJ mol$^{-1}$ [25]. This is comparable to that of the system's rehydration, which provides a driving force for the polymerization reaction.

**Adsorbed amino acids and peptides diffuse on LDH surface.** Despite being adsorbed, amino acids/peptides are never fully immobilized at the LDH surface (Fig. 4a). Their diffusion velocity on the LDH surface is small (5–10 Å ns$^{-1}$) for dehydrated systems (below 5 W/AA), where the amino acids are confined between two layers. In contrast, at higher hydrations, amino acids move rapidly (20–30 Å ns$^{-1}$), exploring the surface of the layer. Remarkably, the mobility of the 40% loaded mixture (MIX2) is only slightly smaller than that of a fully loaded system (MIX1). By studying the amino acids' trajectories on the LDH surface, we observed that their diffusion favors specific directions. Autocorrelation analysis demonstrates that amino acid diffusion follows a six-fold symmetry, templated by the LDH (Supplementary Fig. 6). Diffusion along preferential axes further increases the likelihood of an encounter between two amino acids.

Generally, upon a first binding event, all amino acids spend the majority of time bound to the surface and occasionally desorb (Fig. 4c). Even lysine, which is observed not to be strongly adsorbing, follows this trend upon binding. Importantly, since peptide chains adsorb via their C-termini, peptides mimic the behavior of amino acids; with no strong correlation between chain length, diffusion velocity, and residence time observed for a nascent protein chain (Fig. 4b, d).

## Discussion

In this work we have explored the conceptual challenges associated with the formation of proto-biopolymers on mineral surfaces using the example of oligopeptides and amino acids at LDH surfaces. All of the amino acids tested were observed to be strongly adsorbed on the positively charged LDH surface, with the negative C-terminal oxygens forming H-bonds with the hydroxide groups at the LDH interface. It was notable that, when adsorbed, the molecules were mobile, exploring the full surface of the clay with preferential movement along the six-fold symmetry of the mineral. Amino acids diffused on the surface with a velocity inversely proportional to the crowding at the surface, and occasionally desorbed.

LDH dehydration may arise from wetting—drying cycles, heat, conversion of the water to hydrogen during serpentinization reaction, or as a result of inflow of highly saline water as described in the salt induced peptide formation (SIPF) theory [26]. Dehydration decreases LDH $d$-spacing, consequently increasing amino acid crowding. Even though $d$-spacing is proportional to the number of intercalated atoms, layer undulations are dependent on the nature of the intercalated amino acids. For instance, when amino acids bridge two layers, large static fluctuations are observed. Templated adsorption and partial dehydration create favorable arrangements and environments for the formation of peptide bonds. Remarkably, templating is not dependent on the concentration of amino acids in the interlayer, but rather on the amount of water per unit area of LDH surface.

As with single amino acids, di-, hexa- and 24-mer peptides remain attached by their negative C-terminals, while the backbone desorbs from the surface. This allows the preservation of peptide mobility independently of its length or system concentration. When the peptide sequence features aspartate, the carboxyl side-chain can also adsorb onto the LDH, contributing

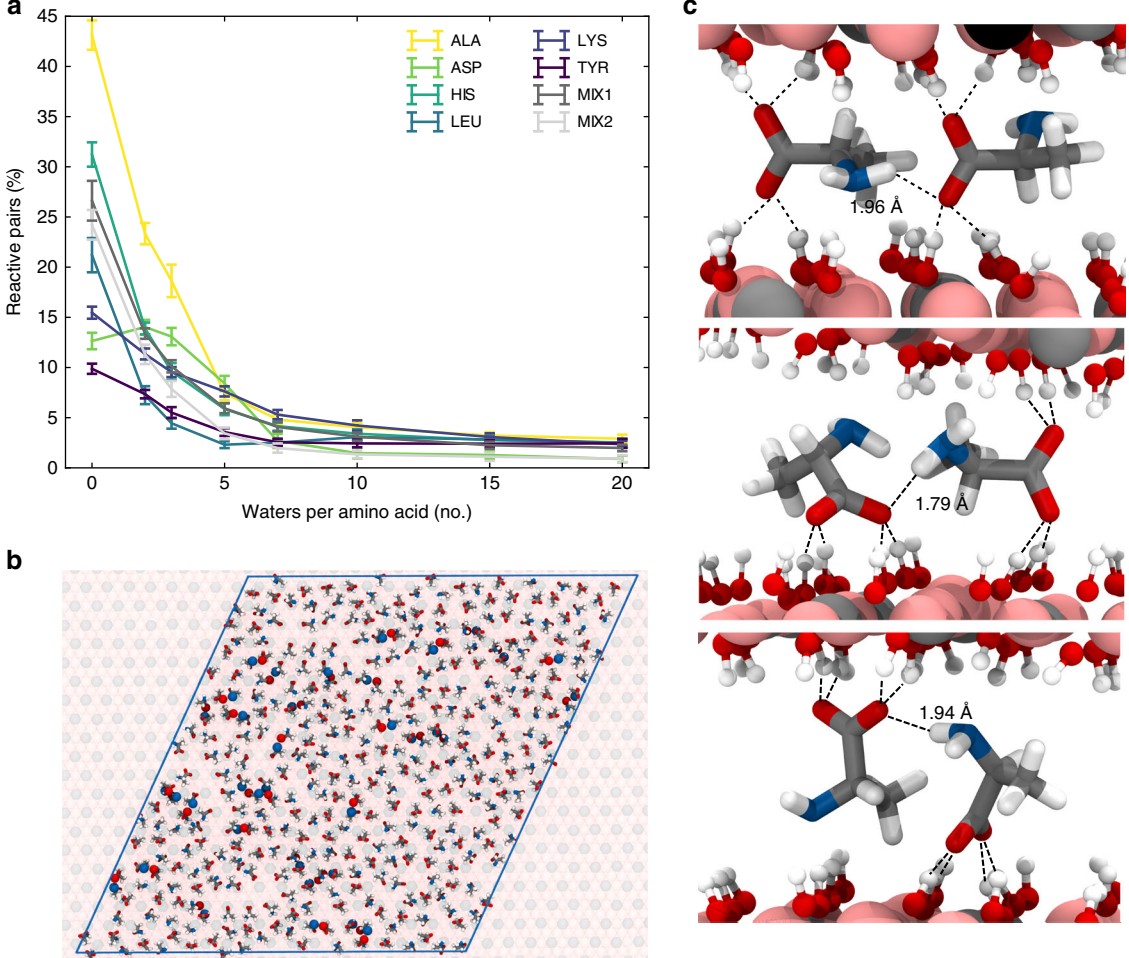

**Fig. 3** Dehydration promotes formation of reactive pairs. **a** Percentage of amino acids' reactive pairs as a function of their hydration (number of water molecules per amino acids). Error bars represent standard deviation. For all amino acids, the percentage of reactive pairs in the system increases upon dehydration. **b** Top view of amino acids adsorbed and co-aligned onto the LDH surface, forming potential reactive pairs (shown spheres, where the backbone nitrogen is colored in blue, and most proximal backbone oxygen in red). **c** Reactive pairs show a range of different possible orientation, involving one or both adjacent LDH layers. Colors are as follows: Mg, pink spheres; Al, gray spheres; Cl, blue spheres; O, red; H, white; N, blue; and backbone is represented with yellow spline. For clarity water and H-atoms on the amino acids are not shown

to the stabilization of the peptide on the surface. Importantly, the model of a realistic system of 24-mers at low concentration features the same adsorption and dynamics of short peptides.

These results suggest a detailed mechanism of peptide formation, as shown in Fig. 4a. At high pH, amino acids adsorb onto the positive LDH layers via their negative C-termini. Partial dehydration creates an energy demand in the system, making condensation reactions increasingly favorable. Formation of a peptide bond leads to the loss of charged group. This, in turn, facilitates the introduction of a new amino acid to an adjacent site, where it then can react with the peptide's C-terminus. The growing peptide chain remains tethered at the LDH surface via the C-terminus of the latest amino acid added.

Our model suggests that, unlike previous observations[27, 28], the formation of a long and biologically relevant peptide is feasible through multiple rehydration cycles and is a slow and controlled process. Figure 5b shows the model of amino acids' polymerization kinetic process (Methods section and Supplementary Fig. 8). The model indicates that in order to form a significant amount of 10-mers (the length of chignolin, the shortest protein), more than 10 rehydration cycles must occur, while the system remains coupled to an infinite amino acid solution bath. When accounting for the timescale of the

emergence of life on the planet, such a process would seem hardly unfeasible.

The LDH-amino acid/peptide coupling can be thought of as a mineral active site, where bond-making between polymers may be facilitated, but the polymer then has no permanent association. Such a mechanism is unlike that previously observed both for nucleic acids adsorbed on LDH[23, 24] and amino acids on clay surfaces[3], while having a strong resemblance to ribosome-catalyzed peptide bond formation[7]. In our simulations we observe that at high concentrations hydrophobic amino acids aggregate. This indicates that, within the clay layers, peptide chains should be able to undergo hydrophobic collapse, an essential mechanism of protein folding. Selectivity dependent on amino acid composition and ordering of peptides may occur due to the slight difference of amino acid affinity to the surface, as well as their relative concentration in the solution bath. In summary, our results outline a testable mechanism for prebiotic peptide formation assisted by LDHs under early Earth conditions.

## Methods

**Molecular models set-up**. This study considers the layered double hydroxide (LDH) of stoichiometry $Mg_3Al(OH)_8$ with one positive charge per unit cell (Supplementary Fig. 1). The LDH layer thickness is 5.3 Å. All intercalated amino

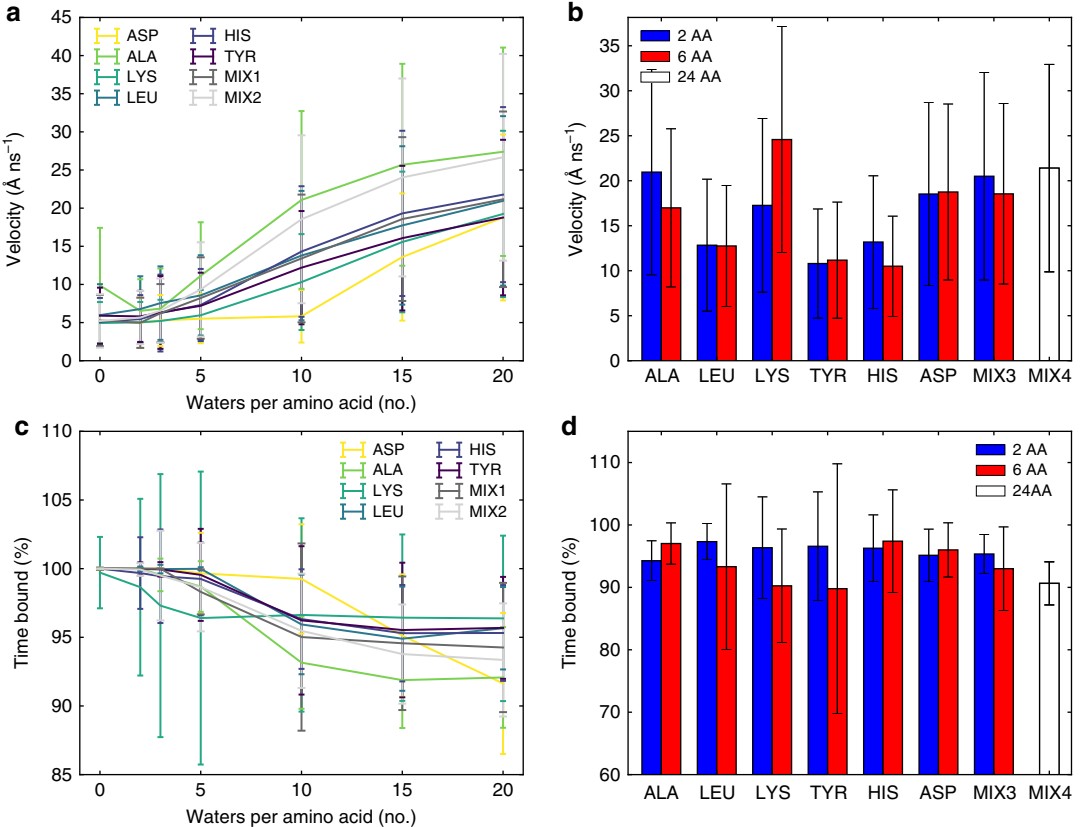

**Fig. 4** Adsorbed amino acids and peptides remain mobile. **a** Velocities of adsorbed amino acids as a function of their hydration (number of water molecules per amino acid). **b** Velocities of adsorbed di-peptides, hexa-peptides, and 24-mers in a fully hydrated system. **c** Time bound of adsorbed amino acids after a first adsorption event as a function of their hydration (number of water molecules per amino acid). **d** Time bound of adsorbed amino acids after a first adsorption event of adsorbed di-peptides, hexa-peptides, and 24-mers in a fully hydrated system. Error bars represent standard deviation. When adsorbed, amino acids and peptides remain mobile upon the LDH surface. Even upon full dehydration, drift on the LDH plane can be observed for all amino acids. Upon a first adsorption event, both amino acids and peptides remain in contact with the LDH surface for majority of the time. Nevertheless, short desorption events (in average ~5% of the time) are still observed

acids' and peptides are deprotonated according to their pKa values to represent pH 9.5. Each group having a pKa lower than 9.5 was deprotonated. Amino acids quantity and charge, as well as the nature and quantity of charge balancing ions used to neutralize each simulation box, are reported in Supplementary Table 1. We considered a range of systems so as to investigate the processes that lie behind peptide formation in the context of the origins of life. The data presented first models a wetting—drying cycle by intercalating a single type of amino acid (ALA, ASP, LEU, LYS, HIS, and TYR) or a mixture (fully counterbalancing layer charge, MIX1, 40% amino acids, and 60% Cl⁻, MIX2) within the hydrated interlayer (20 water per amino acid for all cases but MIX2, where it is 20 water per anion). The water is then removed in a stepwise manner (20, 15, 10, 7, 5, 3, 2, and 0 water).

The behavior of short peptides on the surface of the hydrated LDH was also studied. In this case, a single type of di- (2ALA, 2ASP, 2LEU, 2LYS, 2HIS, 2TYR) or hexa- (6ALA, 6ASP, 6LEU, 6LYS, 6HIS, 6TYR) peptide was intercalated between layers of hydrated LDH. Additionally, realistic mixtures of amino acids on LDH were studied. We created random mixtures based on the percentages of amino acids present in Nature today[29], using the Assemble![30] software. One (MIX3) was built from di- and hexa- peptides at 40% charge balance to LDH, while another (MIX4) was created from three 24-amino-acid-long peptides. Full details of system size and composition are given in Supplementary Table 1.

**Molecular dynamics simulation protocol**. The layered double hydroxide mineral in the simulations was modeled using the ClayFF force field[31]. The force field is specifically parameterized to model clay-like minerals, and the charges were adjusted to create a net +1 charge, as described in our earlier paper[21]. The CHARMM27 force field[32] was used to model the amino acids. ClayFF has been already tested and used with CHARMM force field[33]. Both force fields are parameterized for use along SPC water.

Molecular dynamics simulations were performed with GROMACS 4.6.7[34]. Each simulation was first energy-minimized using the steepest descents algorithm, with convergence when the maximum force on any atom was less than

100 kJ mol⁻¹ nm⁻¹. Then the systems were equilibrated for 0.5 ns in NPT ensemble with velocity-rescale Berendsen thermostat at 300 K, temperature coupling constant set to 0.1 ps, and a semi-isotropic Berendsen barostat at 1 bar, with a pressure coupling constant of 1 ps. The minimization and equilibration simulations were run with real-space particle-mesh-Ewald (PME) electrostatics and a van der Waals cutoff of 1.2 nm. Production runs of 10, 20, or 50 ns (Supplementary Table 1) were then performed. The simulations were run with PME electrostatics and a van der Waals cutoff of 1.4 nm in NPT ensemble, with the same parameters as in the equilibration step illustrated above. For stepwise dehydration simulations, the water was removed after the equilibration phase. The resulting systems were then equilibrated again, prior to the production runs.

**Layer thickness and undulation**. For every equilibrated frame in every dehydration simulation (Supplementary Table 1), all LDH metal atoms coordinates were extracted. In order to assign these coordinates to one of the five simulated clay layers, we exploited the DBSCAN clustering algorithm. As such, layers have been defined as collections of metal atoms being apart by a maximum of 5 Å. Adjacent layers have been identified according to their mean position along the z-axis. d-spacings were calculated by collecting measurements between every pair of adjacent layers (Supplementary Fig. 2a). To obtain information about local d-spacing fluctuations, a 5 × 5 Å sliding window, moved with 1 Å steps on the xy-plane, was applied. Local d-spacing was calculated as the distance between the mean z-axis values of atoms inside the window in two adjacent layers. Mean and standard deviation were calculated for all the collected measures.

To calculate layer undulations (Supplementary Fig. 2b), we translated the center of every layer to the origin, accumulated all metal atoms coordinates, and then calculated the standard deviation of the resulting data sets along the z-axis. The standard deviation of the d-spacing shows the local differences in the thickness of the interlayer. When the standard deviation is small, the layer undulations are correlated; when the standard deviation is large, the layer undulations are de- or anti- correlated.

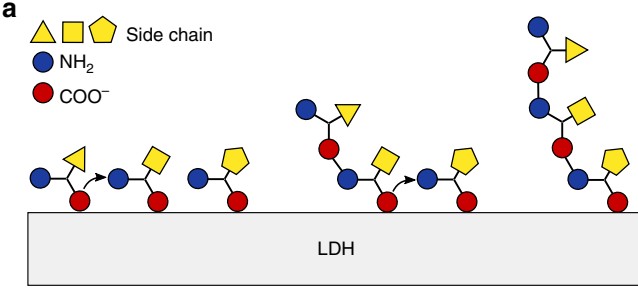

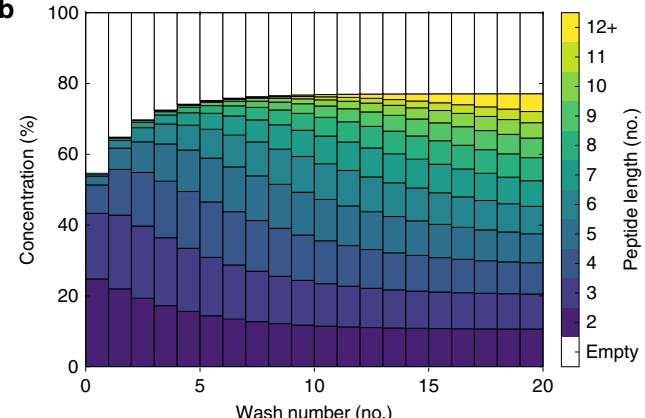

**Fig. 5** Proposed mechanism for LDH supported peptide bond formation. **a** Upon dehydration, the N- and C-termini of adsorbed amino acids co-align, allowing the formation of a peptide bond. The newly formed di-peptide remains tethered via C-terminus only. The bond formation leads to the loss of charge, facilitating introduction of a new amino acid. The N-terminal of amino acid is then able to form a bond with the C-terminal of di-peptide, thus triggering further peptide growth. **b** A kinetic model of peptide growth upon multiple dehydrations—rehydration cycles. After a single dehydration, only dimers to hexamers are observed. Subsequent washing cycles lead to the formation of longer chains

**Amino acid adsorption**. The percentages of adsorbed amino acids were calculated for all dehydration studies and reported as a function of the hydration of the systems (Fig. 2a). For peptide systems (di-peptide, hexapeptide, and mixtures of peptides) the adsorption percentages were also calculated (Fig. 2b). The amino acid/peptide was considered to be adsorbed when a C-terminal oxygen (OT1 and OT2) forms an H-bond with the LDH surface. We adopted a distance cutoff of 2.5 Å, corresponding to the distance of the first hydration layer of LDH. Aspartate can also adsorb via its side-chain (OD1 and OD2), which is reported separately. In the case of di- and hexa- peptides there are twice or six times, respectively, more side-chains than in the backbone and so their adsorbed percentages are scaled accordingly.

**Radial distribution function**. For all the dehydration studies, we computed the radial distribution function (RDF) and the C-terminal atoms of the amino acids using LDH aluminum as reference (Supplementary Fig. 3). The RDF allows us to describe the templating effect of LDH on the arrangement of the amino acids. For comparison, the RDF of aluminum atoms against themselves was also calculated.

**Vectorial analysis**. We calculated the alignment of every amino acid adsorbed with respect to the xy-plane of the LDH surface. A vector was assigned between C and N in the backbone, and its elevation as described in spherical coordinates ($\Theta$ angle) collected. Histograms of elevations were generated with 1° bin size. Angles of 0° identify vectors perpendicular to the plane, and 90° those parallel to the plane (Supplementary Fig. 4).

**Reactive pair count**. For dehydration studies, the likelihood of amino acids co-aligning to form a peptide was calculated (Fig. 3). A reactive pair may be defined as two adsorbed amino acids with their respective C- and N- termini at less than 4 Å distance from one another. In the case of aspartate, alignments that can lead to cyclic reaction were excluded, and the total count of reactive pairs that could lead to either alpha- or beta- peptides is presented.

**Hydration energy**. The energies associated with dehydration of the LDH-amino acid interlayer were calculated as detailed in Wang et al.[35] and are based on the definition of hydration energy introduced in early clay-swelling studies[36],

$$\Delta U_{\mathrm{H}} = \frac{\langle U(N)\rangle - \langle U(0)\rangle}{N},$$ (1)

where $\langle U(N)\rangle$ is the average potential energy of the equilibrium system with N-water molecules and $\langle U(0)\rangle$ is the energy of the fully dehydrated system. The hydration energy can then be compared to the average energy of bulk water ($-33.25$ kJ mol$^{-1}$) and, if lower, the layers will be prone to rehydrate, i.e. swell (Supplementary Fig. 5).

**Diffusion analysis**. For every simulation, the position of the C-terminal carbon is tracked for every equilibrated frame. A carbon was considered as adsorbed if any of its bound oxygens (OT1 and OT2) were within 2.5 Å from the LDH surface. The percentage of time that every carbon spends adsorbed on the surface was calculated after the first adsorption event (Fig. 4c, d). For every carbon adsorbed in two consecutive simulation frames, the diffusion velocity and direction were calculated (Fig. 4a, b). In this situation, we considered diffusion that takes place on the xy-plane only, and its direction was expressed in polar coordinates. All calculated diffusion directions were collected and their distribution represented as histogram. To highlight whether any favorite diffusion axes were present, the direction of the distribution's autocorrelation was calculated (Supplementary Fig. 6). In the case of LDH, the magnesium atoms are hexagonally arranged (i.e. they feature three symmetry axes). If amino acids preferentially diffuse along these symmetry axes, this would be revealed on the autocorrelation plot as six periodic peaks, while random or no diffusion would lead to a flat auto-correlation plot.

**Visualization**. All the snapshots were produced with VMD 1.9.1[37] and graphs were produced with Matplotlib[38].

**Quantum calculations**. In order to calculate the change of partial charges of amino acids upon adsorption on LDH, we have set up five systems—the pure LDH surface of four unit cells, single alanine molecule in vacuum, single alanine adsorbed on the four-unit-cell surface of LDH, four alanine molecules in vacuum and four alanine molecules on an LDH surface. The calculations were performed with CASTEP[39], using norm-conserving planewave pseudopotential and the generalized gradient approximation of Perdew Burke and Ernzerhof[40]. The systems were geometry optimized using density mixing, and the total atomic energy was calculated using the Broyden—Fletcher—Goldfarb—Shanno algorithm; van der Waals forces were applied via the Grimme 06[41]. The following convergence criteria were used for all models: electronic energy tolerance of $1 \times 10^{-6}$ eV, energy change $5 \times 10^{-6}$ eV per atom, maximum displacement of $5 \times 10^{-4}$ Å, and maximum force of $3 \times 10^{-2}$ eV Å$^{-1}$. After geometry optimization, the charge density of the models was analyzed using Hirshfield population analysis.

**Kinetic model**. We developed a kinetic model to describe the process of peptide formation assisted by the LDH interlayer. Negative amino acids and peptides readily adsorb onto the positive LDH surface. Adsorption occurs via the negative C-terminal by activating [reducing the negative charge on] the C-atom for nucleophilic attack (Supplementary Fig. 7):

$$X_1 + ^* \rightarrow X_1^*,$$ (2)

where $X_1$ is the amino acid, * is a surface site and $X_1^*$ is adsorbed/activated amino acid. Upon sufficient dehydration, amino acid can react with another adsorbed amino acid to form a di-peptide $X_2$, or, with an adsorbed peptide $X_n$ and extend it by one monomer unit $X_{n+1}$. The formation of a peptide creates a surface site vacancy:

$$X_1^* + X_n^* \rightarrow X_{n+1}^* + ^*, \text{ where } n = 2, 3, \ldots$$ (3)

Upon rehydration, the formed peptides can desorb, creating another surface site vacancy:

$$X_n^* \rightarrow X_n + ^*.$$ (4)

Since all of the reactions occur only with surface activated species, we omit * for clarity. For all the reactions we assume the same rate constant $k = 1$, and, therefore, it is omitted in the further equations. The following rate equations are identified. The concentration of amino acid $X_1$ will decrease for each peptide bond formation:

$$\frac{\mathrm{d}[X_1]}{\mathrm{d}t} = -\sum_{i=2}^{n} [X_1][X_n].$$ (5)

The concentration of peptide $X_n$ is dependent both on its formation from $X_{n-1}$ and its use in the further polymerization towards $X_{n+1}$ and therefore can be expressed as:

$$\frac{d[X_n]}{dt} = [X_1][X_{n-1}] - [X_1][X_n]; \forall\, n > 1 \tag{6}$$

For the first step, we assume that the LDH layer is fully populated only by amino acids $[X_1] = 100$ and therefore $[X_n] = 0$ for all $n > 1$. Concentrations will converge (i.e. reactions will stop) when $[X_1] = 0$. Upon convergence, we allow the adsorbed species to desorb at 5% to model the species release upon LDH rehydration (Fig. 4). Upon rehydration, LDH vacant sites can be repopulated by the amino acids, bringing the total surface population to 100. We refer to such surface repopulation and peptide release as a wetting step. The process between each wetting step, including of dehydration, peptide formation, rehydration, peptide desorption, and surface repopulation, is referred as a wetting—drying cycle. We track species concentrations after each cycle's convergence, as well as the quantities released from the clay. In Fig. 5a, the concentration after convergence of 20 cycles is presented; a zoom into only 3 cycles is shown in Supplementary Fig. 8a. Supplementary Fig. 8b shows 50 cycles, and Supplementary Fig. 8c shows the cumulant of all species released from LDH. Supplementary Fig. 8d shows cumulative species concentration upon each cycle convergence. The software is developed in Python and the integration is performed with scipy integrate module.

**Data availability**. All other data are available from the authors upon reasonable request.

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

## Acknowledgements

The authors wish to thank AnnMarie O'Donoghue for useful discussions, the Leverhulme Trust (to V.E., D.G.F., and H.C.G.) and EPSRC (to M.T.D. grant ref: EP/P016499/1) for funding, and the Durham HPC Hamilton for the computational resources.

## Author contributions

V.E. and M.T.D. performed the simulations and analyzed the data. V.E., M.T.D., D.G.F., and H.C.G. designed the research and wrote the manuscript.

## Additional information

**Competing interests:** The authors declare no competing financial interests.

