## [Peer Review File · Nature Communications]

REVIEWERS' COMMENTS:

Reviewer #1 (Remarks to the Author):

This is a very interesting modeling study of amino acids condensation on a specific mineral support, namely hydrotalcite. Although the surface condensation of biomolecules has been studied often, we still lack a fundamental understanding that would allow to make sense of disconnected experimental data. The present paper raises some crucial questions and goes some way to answer them. I believe it can be of interest to a general audience, beyond specialists in prebiotic chemistry.

A first original contribution is the evidence for a templating effect of aluminum substitutions in the hydrotalcite lattice, imposing on the amino acids an orientation that favors the condensation of their C-termini. However, amino acids also condense on surfaces that cannot induce this kind of structuring, such as amorphous silica. If the authors' suggestion is correct, condensation should occur more easily on/in hydrotalcites than on silica. Are there experimental results to support this prediction? The authors certainly suggest that there is something special to hydrotalcites: cf. on p.13 "unlike previously observed...(this mechanism has) a strong resemblance to ribosome-catalyzed peptide bond formation". This is a very intriguing idea, but also highly speculative until corroborated experimentally.

The specific treatment of amino acids mobility on p.10 is also an interesting feature. This question is central to the study of bimolecular reactions and is generally overlooked in studies of biomolecules surface condensation. However, the velocity units must be explained – Å nm⁻¹ is not a velocity.

Peristaltic undulations (p.7) of hydrotalcite layers also constitute a rather new observation, even though less related to the central topic of the manuscript.

It should be noted that all of these observations are made possible by the large scale of the modeling, as opposed to other studies of similar systems that only use DFT or similar methods.

The authors have also carried out a valuable and original study of wetting-and-drying cycles that is only cursorily commented in the main text. A separate publication of this part might be justified.

On the down side, I think the paper is less palatable to the general chemist because some questions that (s)he would naturally asked are not treated explicitly. The authors should definitely include in the main text a few sentences to clarify the following questions:

1. How is the layer charge compensated? The "online methods" state that "amino acids... are deprotonated to represent pH 9.5, and in the majority carry a negative charge that counterbalances the positive the positive LDH charge". This is not clear. Was the proportion of deprotonated amino acids chosen on the basis of the corresponding pH? Or of the LDH charge compensation? I assume no other charge compensating ions, such as carbonates, were introduced.

2. Connected to the previous question, what is the acido-basic speciation of amino acids and peptides? They are mostly anions, with one carboxylate and one amine group (cf. "in the majority" above), but are there any zwitterions (or neutral amino acids)? And do proton transfers occur in the various stages of modeling? Also on p.12 "Formation of a peptide bond leads to the loss of a charged group. »: how then is the conservation of charge assured? By the formation of a hydroxide anion?

And on p.8 "the amine side chain is strongly positive" (p. 8): does it mean that it is protonated to an ammonium?

3. What is the criterion for determining through which moiety the amino acids and peptides are adsorbed? (e.g. "adsorption via the backbone" or "adsorption via the side chain" of the Asp molecules.) Is it energetic, based on spatial proximity? Some answers can be found in the supplementary information, but the matter should be clearly stated before the first mention of adsorption mechanism.

Some other points of less general significance:

p. 3, last § « The idea of using hydrated mineral surfaces » - why hydrated ? this is somewhat contradictory with considering "hydrophobic" surfaces

p. 4, 1st §: "Some of these surfaces may have very high enthalpies of hydration, providing a driving force for condensation reaction": This statement should be qualified. Actually, if adsorption of biomonomers occurs from a water solution, the surfaces are already hydrated, and their hydration cannot provide a thermodynamic driving force for condensation.

p.6: a remarkable paucity of simulation data for peptide-mineral interactions:
for peptide-clays, one can mention

Aquino, A. J. A.; Tunega, D.; Gerzabek, M. H.; Lischka, H., Modeling Catalytic Effects of Clay Mineral Surfaces on Peptide Bond Formation. J. Phys. Chem. B 2004, 108, 10120-10130
or even a first attempt, now largely superseded:

Collins, J. R.; Loew, G. H.; Luke, B. T.; White, D. H., Theoretical investigation of the role of clay edges in prebiotic peptide bond formation. Orig. Life Evol. Biosph. 1988, 18, 107-119
Slightly outdated, but addressing an important problem.

p.7, "Irrespective of the identity of the amino acid, the LDH interlayer dehydrates similarly » : I do not understand what exactly the authors mean by « similarly » ; this contention needs to be developed further.

p.10: "peptide bond formation is endergonic with a free energy...comparable to that of the system's rehydration, thus providing a driving force for the polymerization": this sentence must be rephrased. It can be read as meaning that the endergonicity of the reaction provides a driving force, which is contrary to common sense: I suppose the authors actually mean that when it is coupled to the interlayers' rehydration, the global reaction becomes exergonic.

p.11: "Building upon the attractive hypothesis of Russell and Martin ...» : what hypothesis actually ? In the previous text, they are mentioned only as providing a geochemical setting for prebiotic chemistry (alkaline hydrothermal systems), so how is the present work building upon it?

p.12: "...the SIPF theory": it is not so much a theory as a type of reaction, "Salt-Induced Peptide Formation". Give the meaning of the acronym for the general reader.

p.13, "unlike previously observed^{23,24}, » : ref. 24 is missing.

Reviewer #2 (Remarks to the Author):

MANUSCRIPT NUMBER: 144565-0

TITLE: Mineral surface chemistry control for origin of prebiotic peptides

Authors: Valentina Erastova, Matteo T. Degiacomi, Donald Fraser, H. Chris Greenwell

General comments: In my opinion, this paper has important results for the understanding of the adsorption and polymerization of amino acids on mineral surfaces and consequently for the

prebiotic chemistry. However, there is lack between the results of this paper and adsorption/polymerization of amino acids in the context of the prebiotic chemistry. Montmorillonite is one the most studied mineral in prebiotic chemistry. The pH_{pzc} of this clay is about 2.0 meaning that at pH above 2.0, it is negatively charged. Thus, it will adsorb positively charged molecules. The authors used a mineral whose net charge is positive. Thus, it adsorbs molecules negatively charged. Indeed, in general, minerals adsorb charged molecules (D.A.M. Zaia, A review of adsorption of amino acids on minerals: was it important for origin of life? *Amino Acids* 27, 113-118, 2004). In the introduction, the authors should discuss these differences. In addition, because montmorillonite is negatively charged, it has a preference to adsorb amino acids such as lysine, arginine and histidine. However the authors supposed that histidine and lysine adsorbed onto $[Mg_3Al(OH)_8]^+$, it should be noticed that at pH 9.5 lysine has a positive charge from side chain ($pK_{a3} = 10.5$). What is this positive charge effect on the adsorption? The authors could pointed out that in hydrothermal vents pH could reach to pH 11.0 (W. Martin et al., Hydrothermal vents and the origin of life, *Nature Reviews/Microbiology*, 6, 805-814, 2008). However, what is the effect of this high pH on $[Mg_3Al(OH)_8]^+$? Could it be decomposed? Also, was $[Mg_3Al(OH)_8]^+$ a common mineral in prebiotic Earth (R.M. Hazen et al., Mineral evolution, *American Mineralogist*, 93, 1693-1720, 2008)? I also have a few suggestions as below.

Q.1. RESULTS, SECTION Intercalation of amino acids affects LDH layers "Irrespective of the identity of.....charge on amino acids (Figure S2a)".

Comment: Why did not aspartic acid follow this trend?

Q.2. RESULTS, SECTION Amino acids and peptides adsorb on LDHs via their C-terminal, "Here the amine side-chain , reducing the adsorption on the LDH surface".

Comment: If the simulation was carried out at pH higher than pK_a (10.5) of lysine could the adsorption increase?

Q.3 RESULTS, SECTION LDHs promote amino acids polymerization "Alanine shows the highest per unit volume (due its double charge)".

Comment: This result is very interesting for prebiotic chemistry because minerals usually adsorb more amino acids with side-chain charged than amino acids with side-chain uncharged. However, proteins of living being have more amino acids side-chain uncharged than side-chain charged (M.H. Klapper, Independent distribution of amino acids near neighbor pairs into polypeptides. *Biochemistry and Biophysics Research Communications*, 78, 1018-1024, 1977; I.K. Jordan et al., A universal trend of amino acid gain and loss in protein evolution, *Nature*, 433, 633-638, 2005). Besides experiments, simulating the prebiotic Earth or interstellar environments showed high amount of amino acid with uncharged side chain, their adsorption onto mineral is low (D.A.M. Zaia et al., Which amino acids should be used in prebiotic chemistry studies? *Origins of Life and Evolution of the Biosphere* 38, 469-488, 2008). Thus in experiments with wetting/drying cycles could produce peptides with high amount of amino acids like alanine. Thus, the primordial peptides could be more like the proteins of living beings of today. This result could give glue what happen in the prebiotic Earth.

Q.4 Discussion and conclusion

Comment: Reference 24 is cited in the text, but it did not appear in references section

Reviewer #3 (Remarks to the Author):

The submitted manuscript entitled "Mineral Surface Chemistry Control for Origin of Prebiotic

Peptides" is devoted to molecular dynamics (MD) investigation of possible mechanism of peptides synthesis from amino acids within interlayer region of anionic clays – layered double hydroxides (LDHs). The research has scientific novelty and was carried out at a sufficiently high scientific level. Presented results may be of interest to biological community as well as specialists within Earth sciences, chemistry and physics, which makes manuscript a good candidate to be published in Nature Communications. However, the manuscript requires some insignificant corrections and/or additions (as discussed below) before publication.

There are several MD studies devoted to the interaction of amino acids with LDHs (see references below [1-4]), but only studies on the interaction of LDH with nucleic acids / RNA are mentioned by authors in the introduction. In particular, Newman et al. [1] considered the interaction of Phe and Tyr amino acids with Mg₃/Al-LDH, having similar stoichiometry as in manuscript. Kalinichev et al. studied systems with deprotonated Glu anions (1-, 2-) intercalated into Mg₂/Al-LDH [2]. The interaction of anionic amino acids (Asp, Glu) with Mg₂/Al-LDH and the formation of multimolecular hybrid complexes on the LDH surface were investigated in [3]. Interaction / adsorption of cationic Arg amino acid onto Mg₂/Al-LDH surface was studied in [4].

Word "no" in Table S1, meaning "number", should be replaced by "N" or "#" (or something third) for better understanding.

Black arrow (axis) on Fig.S2,c is directed to the right, whereas the number of water molecules per amino acid decreases from 20 to zero. It would be more natural way to arrange snapshots (below arrow) in reverse order or change axis label to "dehydration...".

As a note, in the further development of the proposed idea, it would be interesting to perform MD simulations with an explicit calculation of chemical reactions (peptide bonds formation at different pH, T, hydration, etc. conditions), using, for example, ReaxFF-like approach [5].

References

1. Newman S. P., Cristina T. D., Coveney V. and Jones W. Molecular dynamics simulation of cationic and anionic clays containing amino acids. *Langmuir* 18, 2933–2939 (2002).
2. Kalinichev A. G., Padma Kumar P., and James Kirkpatrick R. (2010). Molecular dynamics computer simulations of the effects of hydrogen bonding on the properties of layered double hydroxides intercalated with organic acids. *Philosophical Magazine*, 90(17-18), 2475-2488.
3. Tsukanov A. A., and Psakhie S. G. (2016). Energy and structure of bonds in the interaction of organic anions with layered double hydroxide nanosheets: A molecular dynamics study. *Scientific reports*, 6, 19986.
4. Tsukanov A. A., and Psakhie S. G. (2016). Adhesion effects within the hard matter–soft matter interface: Molecular dynamics. *Facta Universitatis, Series: Mechanical Engineering*, 14(3), 269-280.
5. Chenoweth K., Van Duin A. C., and Goddard III W. A. (2008). ReaxFF reactive force field for molecular dynamics simulations of hydrocarbon oxidation. *Journal of Physical Chemistry A*, 112(5), 1040-1053.

Response to Reviewers

We greatly appreciate the reviewers' careful scrutiny of our manuscript, and are delighted with their broadly positive in their assessment of the work. We welcome the opportunity to revise the manuscript in light of their suggestions. Below we provide a detailed description of the adjustments made in response to the reviewers' concerns.

Reviewer #1:

A first original contribution is the evidence for a templating effect of aluminum substitutions in the hydrotalcite lattice, imposing on the amino acids an orientation that favors the condensation of their C-termini. However, amino acids also condense on surfaces that cannot induce this kind of structuring, such as amorphous silica. If the authors' suggestion is correct, condensation should occur more easily on/in hydrotalcites than on silica. Are there experimental results to support this prediction? The authors certainly suggest that there is something special to hydrotalcites: cf. on p.13 "unlike previously observed...(this mechanism has) a strong resemblance to ribosome-catalyzed peptide bond formation". This is a very intriguing idea, but also highly speculative until corroborated experimentally.

In this work we show that amino acids polymerization is possible on hydrotalcites. This phenomenon has been already well studied on silicates. The known drawback of silicates is that adsorbed amino acids must feature charged side chains, and that the release of formed peptides is hindered (see ref. 3). In this work we propose a different adsorption mechanism (via the deprotonated C-terminal), enabling adsorption on any type of amino acid. During polymerization, the growing chain remains attached mainly via its C-terminal, making peptide release feasible. A key feature of our proposed mechanism is that long peptide chains should be obtained by multiple repopulation cycles. The aim of this paper is to bring forward this mechanism as a hypothesis to be tested experimentally. Initial research has been undertaken on Aspartate LDHs (unpublished), which showed evidence of small oligomers forming. This work needs to be built on and the results better verified to ensure reproducibility.

The specific treatment of amino acids mobility on p.10 is also an interesting feature. This question is central to the study of bimolecular reactions and is generally overlooked in studies of biomolecules surface condensation. However, the velocity units must be explained – Å nm⁻¹ is not a velocity.

We thank the reviewer for having noted this. This was a typo, velocities were measured in Å ns⁻¹. We have corrected the text accordingly.

How is the layer charge compensated? The "online methods" state that "amino acids... are deprotonated to represent pH 9.5, and in the majority carry a negative charge that counterbalances the positive the positive LDH charge". This is not clear. Was the proportion of deprotonated amino acids chosen on the basis of the corresponding pH? Or of the LDH

charge compensation? I assume no other charge compensating ions, such as carbonates, were introduced. Connected to the previous question, what is the acido-basic speciation of amino acids and peptides? They are mostly anions, with one carboxylate and one amine group (cf. “in the majority” above), but are there any zwitterions (or neutral amino acids)?

We agree that our explanation was not sufficiently clear, and we have provided further details in Methods section. Supplementary Table 1 reports the total charge of each amino acid, and the amount and nature of counterbalancing ions (Cl⁻ or Na⁺) used to neutralize the total charge of each simulation box. We have not used carbonates because in early earth conditions atmospheric carbonate concentrations were much lower than current ones.

In our simulations we deprotonated all groups according to their pKa values. Each group having a pKa lower than 9.5 was deprotonated. For example, at pH 9.5 lysine is zwitterionic, carrying a negative charge on the backbone, and a positive -NH₃⁺ on the side chain.

[...] do proton transfers occur in the various stages of modeling? Also on p.12 “Formation of a peptide bond leads to the loss of a charged group. »: how then is the conservation of charge assured? By the formation of a hydroxide anion?

The reviewer correctly notes that a system containing, e.g., 2 amino acids, may have a different total charge than a system featuring a dipeptide.

We carried out simulation of systems containing single amino acids, peptides and mixtures separately. As such, each system was individually charge-balanced. Full details about the charge of each simulation are provided in Supplementary Table 1.

And on p.8 “the amine side chain is strongly positive” (p. 8): does it mean that it is protonated to an ammonium?

The reviewer is correct, at pH 9.5 lysine will contain an ammonium group on its side chain. We have now corrected this sentence.

What is the criterion for determining through which moiety the amino acids and peptides are adsorbed? (e.g. “adsorption via the backbone” or “adsorption via the side chain” of the Asp molecules.) Is it energetic, based on spatial proximity? Some answers can be found in the supplementary information, but the matter should be clearly stated before the first mention of adsorption mechanism.

Adsorption was determined using a distance cut-off of 2.5 Å, corresponding to the distance of the first hydration layer of the LDH, and is a typical distance for H-bond analysis. We have added this additional information in Methods section, as well as in the caption of Figure 2.

p. 3, last § « The idea of using hydrated mineral surfaces » - why hydrated ? this is somewhat contradictory with considering “hydrophobic” surfaces

Thank you for bringing this to our attention. Hydrophilic mineral surfaces are suitable for

biological catalysis involving polar molecules such as amino acids. We realized that our later mention of hydrophobic/hydrophilic domains may confuse the reader. As this does not provide any information useful to further understand the context of our work, we have decided to remove it.

p. 4, 1st §: “Some of these surfaces may have very high enthalpies of hydration, providing a driving force for condensation reaction”: This statement should be qualified. Actually, if adsorption of biomonomers occurs from a water solution, the surfaces are already hydrated, and their hydration cannot provide a thermodynamic driving force for condensation.

Adsorption of biomonomers does indeed occur from a water solution. Layers with adsorbed species can however subsequently dehydrate because of physical phenomena such as heat or tides. We realize this sentence was unclear, and substituted “hydration” for “rehydration”.

p.6: a remarkable paucity of simulation data for peptide-mineral interactions: for peptide-clays, one can mention

Aquino, A. J. A.; Tunega, D.; Gerzabek, M. H.; Lischka, H., Modeling Catalytic Effects of Clay Mineral Surfaces on Peptide Bond Formation. J. Phys. Chem. B 2004, 108, 10120-10130

or even a first attempt, now largely superseded:

Collins, J. R.; Loew, G. H.; Luke, B. T.; White, D. H., Theoretical investigation of the role of clay edges in prebiotic peptide bond formation. Orig. Life Evol. Biosph. 1988, 18, 107-119 Slightly outdated, but addressing an important problem.

Our wording was poorly chosen, and we have amended it to “*peptide-LDH interactions*”. We thank however the reviewer for having indicated these references, focussing on works about peptides-silicate clays interactions. At p.5, we mention “*Many studies have been carried out on silicate clays to study their potential role in the formation of protobiomolecules*”, and provide four references. The suggested references are very pertinent to this statement, and have therefore decided to add the more recent of the two.

p.7, “Irrespective of the identity of the amino acid, the LDH interlayer dehydrates similarly » : I do not understand what exactly the authors mean by « similarly » ; this contention needs to be developed further.

We have replaced “*similarly*” with “*with the same trend*”. This is then qualified in the second part of the sentence, stating “*indicating that the basal d-spacing is proportional to the number of atoms (organic load) present in the interlayer [...]*”.

p.10: “peptide bond formation is endergonic with a free energy...comparable to that of the system’s rehydration, thus providing a driving force for the polymerization”: this sentence

must be rephrased. It can be read as meaning that the endergonicity of the reaction provides a driving force, which is contrary to common sense: I suppose the authors actually mean that when it is coupled to the interlayers' rehydration, the global reaction becomes exergonic.

The reviewer is right, we are sorry for the confusion. We have rephrased this sentence as follows: *“The formation of a peptide bond releases a molecule of water, thus contributing to the rehydration of the interlayer. We note that peptide bond formation is endergonic with free energy change of 10-20 kJ mol⁻¹. This is comparable to that of system's rehydration, that provides a driving force for the polymerization reaction.”*

p.11: “Building upon the attractive hypothesis of Russell and Martin ...» : what hypothesis actually ? In the previous text, they are mentioned only as providing a geochemical setting for prebiotic chemistry (alkaline hydrothermal systems), so how is the present work building upon it?

Russell and Martin suggest that alkaline hydrothermal vents, and the mineral assemblages and microstructures found there, may provide a geochemical environment where prebiotic chemistry may have been favoured. We agree with the reviewer that the link between Russell and Martin's work and our own is not as direct as this statement reads. Therefore we have removed an explicit mention of their work in Discussion.

p.12: “...the SIPF theory”: it is not so much a theory as a type of reaction, “Salt-Induced Peptide Formation”. Give the meaning of the acronym for the general reader.

We have now added the definition of this acronym.

p.13, “unlike previously observed^{23,24}, » : ref. 24 is missing.

We are sorry for this oversight; the references have now been updated.

Reviewer #2:

[...] there is lack between the results of this paper and adsorption/polymerization of amino acids in the context of the prebiotic chemistry. Montmorillonite is one the most studied mineral in prebiotic chemistry. The pHPzc of this clay is about 2.0 meaning that at pH above 2.0, it is negatively charged. Thus, it will adsorb positively charged molecules. The authors used a mineral whose net charge is positive. Thus, it adsorbs molecules negatively charged. Indeed, in general, minerals adsorb charged molecules (D.A.M. Zaia, A review of adsorption of amino acids on minerals: was it important for origin of life? Amino Acids 27, 113-118, 2004). In the introduction, the authors should discuss these differences.

We thank the reviewer for the suggested reference; we have added it to the main text. Unfortunately size limitations in the introduction section did not allow us to further discuss this point without sacrificing others.

In addition, because montmorillonite is negatively charged, it has a preference to adsorb amino acids such as lysine, arginine and histidine. However the authors supposed that histidine and lysine adsorbed onto $[Mg_3Al(OH)_8]^+$, it should be noticed that at pH 9.5 lysine has a positive charge from side chain ($pK_{a3} = 10.5$). What is this positive charge effect on the adsorption?

Although upon dehydration lysine does adsorb on the LDH surface, it does so to a lesser extent than all other negatively charged amino acids (see Figure 2a). Lysine adsorbs via its backbone as all other amino acids, and when adsorbed it follows their same behaviour in terms of adsorption times and velocities (see Figure 4).

The authors could pointed out that in hydrothermal vents pH could reach to pH 11.0 (W. Martin et al., Hydrothermal vents and the origin of life, Nature Reviews/Microbiology, 6, 805-814, 2008). However, what is the effect of this high pH on $[Mg_3Al(OH)_8]^+$? Could it be decomposed?

LDH are synthesized at high pH (>8), and are stable even at extremely high pH (>12). See for instance:

- J. W. Boclair and P. S. Braterman, "Layered Double Hydroxide Stability. 1. Relative Stabilities of Layered Double Hydroxides and Their Simple Counterparts", Chem. Mater., 1999
- Seron and F. Delorme, "Synthesis of layered double hydroxides (LDHs) with varying pH: A valuable contribution to the study of Mg/Al LDH formation mechanism", Journal of Physics and Chemistry of Solids, 2008

This property enables them to exist in hydrothermal vents.

Also, was $[Mg_3Al(OH)_8]^+$ a common mineral in prebiotic Earth (R.M. Hazen et al., Mineral

evolution, American Mineralogist, 93, 1693-1720, 2008)?

We thank the reviewer for having pointed this out. We have added this reference in the introduction, along with a mention that such surface was indeed common in early earth.

Q.1. RESULTS, SECTION Intercalation of amino acids affects LDH layers “Irrespective of the identity of.....charge on amino acids (Figure S2a)”. Comment: Why did not aspartic acid follow this trend?

Aspartate is deprotonated at both side chain and backbone, leading to a total charge of -2. For this reason, to compensate the LDH charge, for aspartate systems we have used half the concentration of all other amino acids tested in this work. As a consequence, the number of atoms intercalated in the interlayer was smaller in aspartate systems, leading to smaller *d*-spacings (Supplementary Figure 2a). We should also notice that the strong negative charge of aspartate makes it more adsorbing than all other amino acids, as shown in Figure 2.

Q.2. RESULTS, SECTION Amino acids and peptides adsorb on LDHs via their C-terminal, “Here the amine side-chain , reducing the adsorption on the LDH surface”. Comment: If the simulation was carried out at pH higher than pKa (10.5) of lysine could the adsorption increase?

At such a pH the lysine side chain would deprotonate. On the basis of results for other amino acids, we expect that lysine adsorption should increase as a result.

Q.3 RESULTS, SECTION LDHs promote amino acids polymerization “Alanine shows the highest per unit volume (due its double charge)”. Comment: This result is very interesting for prebiotic chemistry because minerals usually adsorb more amino acids with side-chain charged than amino acids with side-chain uncharged. However, proteins of living being have more amino acids side-chain uncharged than side-chain charged (M.H. Klapper, Independent distribution of amino acids near neighbor pairs into polypeptides. Biochemistry and Biophysics Research Communications, 78, 1018-1024, 1977; I.K. Jordan et al., A universal trend of amino acid gain and loss in protein evolution, Nature, 433, 633-638, 2005). Besides experiments, simulating the prebiotic Earth or interstellar environments showed high amount of amino acid with uncharged side chain, their adsorption onto mineral is low (D.A.M. Zaia et al., Which amino acids should be used in prebiotic chemistry studies? Origins of Life and Evolution of the Biosphere 38, 469-488, 2008). Thus in experiments with wetting/drying cycles could produce peptides with high amount of amino acids like alanine. Thus, the primordial peptides could be more like the proteins of living beings of today. This result could give glue what happen in the prebiotic Earth.

Indeed, in this work we look at the interactions in Early Earth conditions (hydrothermal vent-like, high pH). In these conditions amino acids adsorb via their deprotonated carboxylic group of the backbone on the positive LDHs. So, amino acids with neutral side chain will readily adsorb and form reactive pairs. This in principle allows the uptake of any amino acid from the

environment. Therefore the composition of the peptides produced via the method suggested in this work will be primarily dictated by the availability of the amino acids. This is not the case for negative silicate clays such as montmorillonite, where the amino acids adsorb mainly via their charged side chains.

Q.4 Discussion and conclusion. Comment: Reference 24 is cited in the text, but it did not appear in references section

We are sorry for this oversight; the references have now been updated.

Reviewer #3:

There are several MD studies devoted to the interaction of amino acids with LDHs (see references below [1-4]), but only studies on the interaction of LDH with nucleic acids / RNA are mentioned by authors in the introduction. In particular, Newman et al. [1] considered the interaction of Phe and Tyr amino acids with Mg₃/Al-LDH, having similar stoichiometry as in manuscript. Kalinichev et al. studied systems with deprotonated Glu anions (1-, 2-) intercalated into Mg₂/Al-LDH [2]. The interaction of anionic amino acids (Asp, Glu) with Mg₂/Al-LDH and the formation of multimolecular hybrid complexes on the LDH surface were investigated in [3]. Interaction / adsorption of cationic Arg amino acid onto Mg₂/Al-LDH surface was studied in [4]. References:

- 1. Newman S. P., Cristina T. D., Coveney V. and Jones W. Molecular dynamics simulation of cationic and anionic clays containing amino acids. Langmuir 18, 2933–2939 (2002).*
- 2. Kalinichev A. G., Padma Kumar P., and James Kirkpatrick R. (2010). Molecular dynamics computer simulations of the effects of hydrogen bonding on the properties of layered double hydroxides intercalated with organic acids. Philosophical Magazine, 90(17-18), 2475-2488.*
- 3. Tsukanov A. A., and Psakhie S. G. (2016). Energy and structure of bonds in the interaction of organic anions with layered double hydroxide nanosheets: A molecular dynamics study. Scientific reports, 6, 19986.*
- 4. Tsukanov A. A., and Psakhie S. G. (2016). Adhesion effects within the hard matter–soft matter interface: Molecular dynamics. Facta Universitatis, Series: Mechanical Engineering, 14(3), 269-280.*
- 5. Chenoweth K., Van Duin A. C., and Goddard III W. A. (2008). ReaxFF reactive force field for molecular dynamics simulations of hydrocarbon oxidation. Journal of Physical Chemistry A, 112(5), 1040-1053.*

We thank the reviewer for having pointed these references out, and have added the second suggested reference in the introduction. Although the other references are pertinent in the context of amino acid adsorption on mineral surfaces, we have chosen not to add them as they are not addressing the topic of formation of proto-biomolecules or origins of life.

Word “no” in Table S1, meaning “number”, should be replaced by “N” or “#” (or something third) for better understanding.

We have now substituted “No” with “#”.

Black arrow (axis) on Fig.S2,c is directed to the right, whereas the number of water molecules per amino acid decreases from 20 to zero. It would be more natural way to arrange snapshots (below arrow) in reverse order or change axis label to “dehydration...”.

We agree with the reviewer, the direction of the arrow could have confused the reader. We have relabelled the figure to clarify our intent, i.e. that the arrow indicates the direction of our simulation protocol, stepwise dehydrating the interlayers.

As a note, in the further development of the proposed idea, it would be interesting to perform MD simulations with an explicit calculation of chemical reactions (peptide bonds formation at different pH, T, hydration, etc. conditions), using, for example, ReaxFF-like approach [5].

We thank the reviewer for the suggestion. Indeed, observing peptide bonds formation in simulation and testing their dependence on different conditions would definitely be a very exciting project continuation. We have recently performed preliminary testing of ReaxFF for our systems, though we are at a too early stage to draw any conclusion. Using QM/MM methods would be suitable for such a study, and should be considered for future modelling work.